# Decoupling General and Personalized Knowledge in Federated Learning via Additive and Low-rank Decomposition

## ABSTRACT

To address data heterogeneity, the key strategy of Personalized Federated Learning (PFL) is to decouple general knowledge (shared among clients) and client-specific knowledge, as the latter can have a negative impact on collaboration if not removed. Existing PFL methods primarily adopt a parameter partitioning approach, where the parameters of a model are designated as one of two types: parameters shared with other clients to extract general knowledge and parameters retained locally to learn client-specific knowledge. However, as these two types of parameters are put together like a jigsaw puzzle into a single model during the training process, each parameter may simultaneously absorb both general and client-specific knowledge, thus struggling to separate the two types of knowledge effectively. In this paper, we introduce FedDecomp, a simple but effective PFL paradigm that employs parameter additive decomposition to address this issue. Instead of assigning each parameter of a model as either a shared or personalized one, FedDecomp decomposes each parameter into the sum of two parameters: a shared one and a personalized one, thus achieving a more thorough decoupling of shared and personalized knowledge compared to the parameter partitioning method. In addition, as we find that retaining local knowledge of specific clients requires much lower model capacity compared with general knowledge across all clients, we let the matrix containing personalized parameters be low rank during the training process. Moreover, a new alternating training strategy is proposed to further improve the performance. Experimental results across multiple datasets and varying degrees of data heterogeneity demonstrate that FedDecomp outperforms state-of-the-art methods up to 4.9%.

## CCS CONCEPTS

• **Computing methodologies → Distributed artificial intelligence**.

## KEYWORDS

Personalized Federated Learning, Data Heterogeneity, Parameter Decomposition

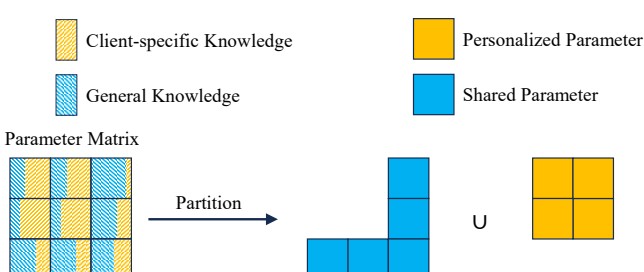

**Figure 1: A toy example to illustrate the partition based method.**

## 1 INTRODUCTION

Federated learning (FL) [28] allows clients to collaboratively train a global model without directly sharing their raw data. It has garnered widespread attention in the design of multimedia artificial intelligence systems [4, 20, 36, 41]. A central challenge in FL is data heterogeneity, where the data distributions across diverse clients are not independently and identically distributed (non-IID). Such disparities in data distributions hamper the training of the global model, leading to a decrease in the performance of FL [10, 22, 39].

To confront this challenge, the concept of Personalized Federated Learning (PFL) has been introduced. Within PFL studies, it is widely accepted that the knowledge learned by a client should be decoupled into two categories: the general knowledge across all clients and client-specific knowledge for this client [6, 34, 38]. The former is used for sharing among clients to promote collaboration, while the latter is retained locally to keep personalization and reduce the impact of data heterogeneity on collaboration. This understanding prompts mainstream PFL research to propose a partition based method where each parameter in the client's personalized model is designated as one of two types before training begins: parameter shared with other clients to extract general knowledge and parameter retained in personalization to learn client-specific knowledge. A multitude of studies have emerged. For instance, FedPer [2] focuses on personalizing the classifier, whereas FedBN [24] targets the personalization of Batch Normalization layers. FedCAC [34] proposes to select personalized parameters based on measurable metrics.

Although the aforementioned methods have achieved some success and attracted widespread attention, they still haven't effectively separated the two types of knowledge. This is because the personalized parameters and shared parameters of a model learn from local data together as a whole, allowing each parameter to potentially absorb both general and client-specific knowledge simultaneously. This means that personalized parameters may contain some general knowledge that should be shared, and shared parameters may contain some client-specific knowledge that should be personalized. Fig. 1 is a toy example that illustrates this point intuitively. The square on the far left represents a personalized model parameter

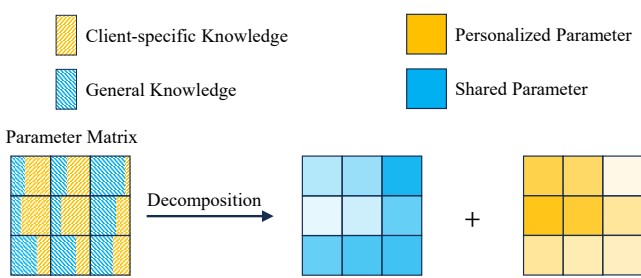

**Figure 2: A toy example to illustrate our method. The depth of blue/orange in the shared/personalized parameters indicates the amount of knowledge from the corresponding parameters in the original parameter matrix.**

matrix containing nine parameters. The blue diagonal rectangle and the orange diagonal rectangle within each parameter represent the general knowledge and personalized knowledge contained in that parameter, respectively, with their area size indicating the amount of knowledge. It can be seen that each parameter contains both general and personalized knowledge. The partition based method divides some parameters into shared parameters (as shown by the blue squares), and the rest into personalized parameters (as shown by the orange squares). Obviously, this "black or white" partitioning method cannot achieve knowledge decoupling within parameters, leading to some client-specific knowledge in shared parameters being shared, thereby reducing the degree of model personalization; while some shared knowledge in personalized parameters is personalized, thus reducing the level of client collaboration.

In this paper, we propose a new PFL paradigm based on parameter additive decomposition, called FedDecomp, to address the aforementioned issues. Unlike methods based on partitioning, which classify a parameter as either personalized or shared, FedDecomp decomposes each parameter into the sum of two parameters before training begins: one shared to facilitate knowledge exchange among clients and one retained locally to maintain personalization. Furthermore, we find that in PFL, general knowledge should be retained in shared parameters with high model capacity to cover all clients, while a client's specific knowledge can be learned by personalized parameters with lower model capacity as a supplement to general knowledge. Hence, FedDecomp constrains the matrix containing personalized parameters to be low rank. This allows the personalized part to focus its learning on the most critical aspects of the local knowledge and reduce the overfitting to the local knowledge. Consequently, it helps to retain a significant portion of the general knowledge acquired from other clients, thereby enhancing generalization. Fig. 2 illustrates our proposed method. It can be seen that in our method, both the general knowledge and client-specific knowledge in each parameter can be decoupled and accordingly captured by shared and personalized parameters, achieving more efficient client collaboration and personalization.

In addition, different from the current methods which simply train personalized parameters and shared parameters simultaneously, we examine the training order of shared and personalized parameter matrices during local updates. Specifically, we propose to initially train the personalized low-rank part to mitigate the

influence of non-IID data, followed by training the shared full-rank part. Our findings suggest that adopting an alternating approach, unlike concurrent training methods, yields greater benefits.

Our primary contribution in this paper can be summarized as follows:

- We introduce a new method of decomposing shared and personalized parameters in PFL, namely FedDecomp. Specifically, we decompose each layer of the personalized model into the sum of a shared full-rank part to preserve general knowledge and a personalized low-rank part to preserve client-specific knowledge.
- We introduce an innovative training strategy designed to optimize FedDecomp, effectively mitigating the implications of non-IID data and significantly boosting performance.
- We evaluate FedDecomp across multiple datasets and under varied non-IID conditions. Our findings underscore the efficacy of the FedDecomp method we propose.

## 2 RELATED WORK

PFL has emerged as a prevalent research direction to handle the non-IID problem in FL. Current PFL methods can be mainly divided into meta-learning-based methods [1, 8], fine-tuning-based methods [5, 17], clustering-based methods [3, 31], model-regularization-based methods [23, 33], personalized-aggregation-based methods [14, 40], and parameter-partition-based methods. Among these methods, the parameter-partition-based method has attracted a lot of attention due to its simplicity and effectiveness.

**Parameter-partition-based method.** The core idea of this kind of method is to share part of the original model's parameters while personalizing the other part. Representative works include selecting specific layers for personalization, such as FedPer [2], FedRep [7], and GPFL [38] proposing to personalize classifiers. FedBN [24] and MTFL [29] suggest to personalize the Batch Normalization (BN) layers. LG-FedAvg [25] and FedGH [37] propose to personalize feature extractor. Other works employ Deep Reinforcement Learning (DRL) or hypernetworks technologies to automate the selection of specific layers for personalization [27, 32]. Still, some other research no longer selects personalized parameters based on layers but on each individual parameter, making more fine-grained choices to personalize parameters sensitive to non-IID data [34]. In recent years, some studies propose another kind of personalized parameter partitioning method. Unlike the previous method, the core idea of this method is to add additional personalized layers to the original model. For example, ChannelFed [42] introduces a personalized attention layer to redistribute weights for different channels in a personalized manner. [30] proposes to add a bottleneck module for personalization after each feedforward layer.

**Parameter-decomposition-based method.** A few PFL works also utilize parameter decomposition techniques. For instance, Fedpara [15] decomposes the personalized model parameter matrix into the Hadamard product of two low-rank matrices. It has been proven that the parameter matrix after the Hadamard product still possesses a high rank, thus not sacrificing model capacity. This way, it only requires uploading the low-rank matrix with fewer parameters during training, thereby reducing communication overhead. Factorized-FL [16] decomposes the model parameter matrix into the product of a column vector and a row vector. During training, it

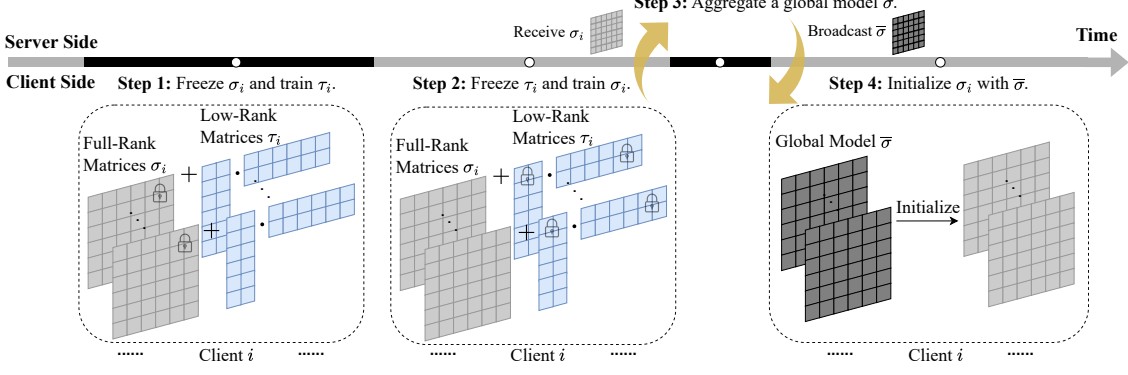

**Figure 3: Overview of one client in FedDecomp in one communication round.**

personalizes the row vector while sharing the column vector. This is essentially a low-rank decomposition technique aimed at reducing communication overhead by only uploading the column vector. FedSLR [13], on the other hand, performs a low-rank decomposition of the parameter matrix when the server distributes the model, thus reducing the communication overhead of the downlink.

It is evident that our approach differs significantly from the current decomposition-based methods, from objectives to methodologies. The current methods mainly focus on the communication issue in PFL by reducing the amount of communication between clients and the server through low-rank decomposition of parameters. Our paper, however, focuses on the decoupling and extraction of knowledge in PFL. Through additive decomposition, it decouples the learning of general knowledge and client-specific knowledge. By constraining the personalized matrix to be low-rank, it coordinates the relationship between general and client-specific knowledge.

## 3 METHOD

### 3.1 Overview of FedDecomp

We first give an overview of FedDecomp. As illustrated in Fig. 3, each layer of client $i$'s personalized model is decomposed into the sum of a full-rank matrix and a low-rank matrix. The training process in each communication round can be summarized as follows: 1) each client $i$ freezes its full-rank matrices $\sigma_i$ and updates the low-rank matrices $\tau_i$. 2) Then, each client $i$ turns to update $\sigma_i$ and freeze $\tau_i$. After local updating, all clients upload the full-rank part to the server while keeping the low-rank part private. 3) The server receives clients' full-rank matrices and aggregates them to generate a global model $\overline{\sigma}$. After doing this, the server sends $\overline{\sigma}$ back to all clients. 4) Each client receives the global model and uses it to initialize the full-rank matrices.

### 3.2 Problem Definition of PFL

PFL, in contrast to traditional FL algorithms that train a general model for all clients, strives to develop a personalized model for each client $i$, denoted as $w_i$, specializing in capturing the unique characteristics of its local data distribution $D_i$. In recent PFL research, there is a consensus that the knowledge acquired by individual clients comprises both general knowledge and client-specific

knowledge. In non-IID scenarios, since different clients have distinct data distributions (i.e., $D_i \neq D_j, i \neq j$), it is difficult to extract general knowledge and thus brings challenges to client collaboration. To address this problem, PFL decouples $w_i$ into a shared part $\sigma$ and a personalized part $\tau_i$ to learn general knowledge and client-specific knowledge respectively. Formally, the training objective can be formulated as

$$\min_{\sigma, \tau_1, \tau_2, \ldots \tau_N} \sum_{i=1}^{N} L_i(\sigma, \tau_i; D_i), \quad (1)$$

where $L_i(\sigma; \tau_i; D_i)$ denotes the loss function of client $i$ and $N$ is the total number of clients. To optimize the target function in Eq. (1), recent studies have put forth various PFL methods to partition $\tau_i$ and $\sigma$. While these endeavors have shown promise, the question of how to further refine the decomposition of these two parameter components still presents an unresolved challenge.

### 3.3 Low-rank Parameter Decomposition

We observe that shared parameters responsible for extracting general knowledge benefit from a high model capacity. In contrast, personalized parameters are tasked with learning knowledge that complements the general understanding for specific local tasks (i.e., client-specific knowledge), therefore, it is sufficient to use a low-rank matrix to represent these personalized parameters. Based on this observation, we propose FedDecomp, an additive low-rank decomposition technique. Details about this method are as follows.

**Additive Decomposition of Personalized Models:** Assume that each personalized model has a set of weights $\boldsymbol{\theta_i} = \{\theta_i^k\}_{k=1}^{L}$, where $\theta_i^k$ is the weights for the $k$-th layer and $L$ is the total layer number. Each weight matrix $\theta_i^k$ is originally full-rank. In FedDecomp, we decompose $\theta_i^k$ as

$$\theta_i^k = \sigma_i^k + \tau_i^k, k \in [1, L], \quad (2)$$

where $\sigma_i^k, k \in [1, L]$ is a full-rank parameter matrix that is shared across all clients, and $\tau_i^k, k \in [1, L]$ is a personalized low-rank parameter matrix. In the following, we employ the notation $\boldsymbol{\theta_i}$, $\boldsymbol{\sigma_i}$ and $\boldsymbol{\tau_i}$ to denote the complete model parameter set, the full-rank parameter set, and the low-rank parameter set specific to client $i$,

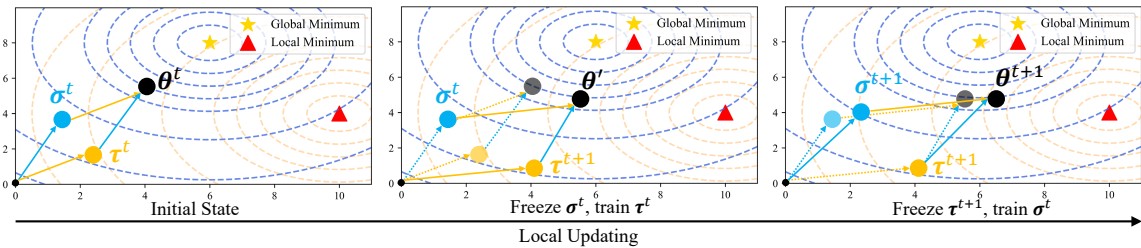

Figure 4: A toy example to illustrate the alternating training in FedDecomp.

respectively. Additionally, we use $\theta_i^k$, $\sigma_i^k$, and $\tau_i^k$ to represent the parameter matrices for layer $k$ within client $i$.

Next, we present the methods for imposing low-rank constraints on $\tau_i$.

**Low-rank Decomposition of Fully-Connected Layers:** For fully-connected layers, the dimension of $\tau_i^k$ is $I \times O$, where $I$ and $O$ represent the input and output dimensions. We constrain $\tau_i^k$ through a low-rank decomposition as follows:

$$\tau_i^k = B_i^k A_i^k, \quad (3)$$

where $B_i^k \in \mathbb{R}^{I \times (R_l \cdot \min(I,O))}$ and $A_i^k \in \mathbb{R}^{(R_l \cdot \min(I,O)) \times O}$.

The $R_l$ serves as a hyper-parameter designed to regulate the rank of $\tau_i^k$ within fully-connected layers. Its value falls within the range of $0 < R_l \le 1$.

**Low-rank Decomposition of Convolutional Layers:** In contrast to fully-connected layers, convolutional layers involve multiple kernels, resulting in $\tau_i^k \in I \times O \times K \times K$ dimensions. However, we can still apply a low-rank decomposition to constrain $\tau_i^k$ as follows:

$$\tau_i^{k*} = B_i^k A_i^k \in \mathbb{R}^{(I \cdot K) \times (O \cdot K)}, \quad (4)$$

$$B_i^k \in \mathbb{R}^{(I \cdot K) \times (R_c \cdot \min(I,O) \cdot K)} \text{ and } A_i^k \in \mathbb{R}^{(R_c \cdot \min(I,O) \cdot K) \times (O \cdot K)},$$

$$\tau_i^k = \text{Reshape}(\tau_i^{k*}) \in \mathbb{R}^{I \times O \times K \times K}.$$

The $R_c$ is a hyper-parameter used to control the rank of $\tau_i^k$ within convolutional layers. Its value is within the range of $0 < R_c \le 1$.

During training, both $B$ and $A$ serve as trainable parameter matrices. We initialize $A$ with random Gaussian values and $B$ with zeros, which means $\tau_i^k$ starts as zero at the beginning of training.

The hyper-parameters $R_l$ and $R_c$ play crucial roles in controlling the rank of parameters within fully connected and convolutional layers, respectively. As the rank increases, the learning capacity of personalized parameters within the model gradually improves. However, if the rank is set too low, $\tau_i$ may struggle to effectively capture client-specific knowledge, making $\sigma_i$ highly susceptible to non-IID data distributions. This, in turn, negatively impacts collaboration among clients. In contrast, if the rank is too large, $\tau_i$ may start to absorb some of the general knowledge that should be learned by $\sigma_i$, diminishing the level of collaboration among clients. For simplicity, in the FedDecomp approach, we apply the same $R_c$ to all convolutional layers and the same $R_l$ to all fully-connected layers. This simplification streamlines the model architecture and hyper-parameter tuning process.

### 3.4 Coordinate Training Between $\sigma$ and $\tau$

To better extract general knowledge, in contrast to the common practice where personalized and shared parameters are trained simultaneously, we find that a more effective strategy is to initially train the low-rank parameters. This alternating approach helps mitigate the impact of non-IID data before proceeding to train the full-rank parameters. Formally, in each communication round $t \in [1, T]$, we first optimize the low-rank parameters $\tau_i$ for $E_{\text{lora}}$ epochs by

$$\tau_i^{t+1} = \arg\min_{\tau_i} L_i(\tau_i^t, \sigma_i^t; D_i). \quad (5)$$

Then optimize the full-rank parameters $\sigma_i$ for $E_{\text{global}}$ epochs by

$$\sigma_i^{t+1} = \arg\min_{\sigma_i} L_i(\tau_i^{t+1}, \sigma_i^t; D_i). \quad (6)$$

We set $E_{\text{lora}} + E_{\text{global}} = E$, where $E$ is the total number of local update epochs in one round. These hyper-parameters play an important role in balancing the learning dynamics between two key components, $\sigma_i$ and $\tau_i$. When $E_{\text{lora}}$ is set higher, it results in $\sigma_i$ learning less knowledge. Consequently, the degree of knowledge sharing among clients diminishes. In contrast, if $E_{\text{lora}}$ is set too low, $\sigma_i$ ends up acquiring a significant amount of client-specific knowledge. This scenario increases the risk of clients sharing knowledge that is more susceptible to non-IID data. In special cases, when $E_{\text{lora}} = 0$, the FedDecomp framework degenerates into FedAvg. Similarly, when $E_{\text{global}} = 0$, FedDecomp transforms into local training with low-rank parameters, without any collaborative efforts among clients.

After local updating, each client $i$ uploads $\sigma_i^{t+1}$ to the server while keeping the $\tau_i^{t+1}$ private. The server computes a global model $\overline{\sigma}^{t+1}$ by aggregating all clients' $\sigma_i^{t+1}$ through

$$\overline{\sigma}^{t+1} = \frac{1}{N} \sum_{i=1}^{N} \sigma_i^{t+1}, \quad (7)$$

and sends it back to clients. The detailed training process is summarized in the Algorithm 1.

To explain our intuition for proposing alternating training, we employ a toy example to illustrate the local update phase of each client's personalized model within the parameter space. As shown in Fig. 4, the yellow ★ and red △ denote the optimum points of the global model on all clients' data (global loss minimum point) and the personalized model on the client's data (local loss minimum point), respectively. Under the influence of non-IID, there is a big difference between global knowledge and local knowledge of clients. This makes the local minimum point far away from the global minimum

---

**Algorithm 1** FedDecomp

---

**Input:** Each client's initial personalized parameter matrices $\tau_i^1$; The global shared parameter matrices $\overline{\sigma}^1$; Number of clients $N$; Total communication round $T$; Global matrices update epoch number $E_{\text{global}}$; Low-rank matrices update epoch number $E_{\text{lora}}$ ;

**Output:** Personalized model parameter matrices $\theta_i^T$ for each client.

**for** $t = 1$ to $T$ **do**

    **Client-side:**

    **for** $i = 1$ to $N$ **in parallel do**

        Initializing $\sigma_i^t$ with $\overline{\sigma}^t$.

        Updating $\tau_i^t$ by (5) for $E_{\text{lora}}$ epochs to obtain $\tau_i^{t+1}$.

        Updating $\sigma_i^t$ by (6) for $E_{\text{global}}$ epochs to obtain $\sigma_i^{t+1}$.

        Sending $\sigma_i^{t+1}$ to the server.

    **end for**

    **Server-side:**

    Aggregating a global model $\overline{\sigma}^{t+1}$ by (7).

    Sending $\overline{\sigma}^{t+1}$ to each client $i$.

**end for**

---

point. The client's personalized model $\theta$ is decomposed into the sum of a shared part $\sigma$ and a personalized part $\tau$. Since we first train the personalized part, the client-specific knowledge is mostly learned by $\tau$ and the shift of $\theta$ to the local minimum point is mainly done by $\tau$. Therefore, when training $\sigma$, it moves less towards the local minimum point (i.e., less affected by non-IID data), so it can better extract general knowledge. In Section 4.4, we conduct an experiment to further validate this intuition.

### 3.5 Training Cost Analysis

In this section, we analyze the memory usage, computation cost, and communication cost of FedDecomp in each client $i$ compared to the baseline method FedAvg.

**Memory usage:** FedAvg needs to maintain a set of full-rank parameter set $\theta_i$. According to Eq. (2), Eq. (3) and Eq. (4), FedDecomp needs to maintain a full-rank parameter set $\sigma_i$ whose number of trainable parameters is equivalent to $\theta_i$, and a low-rank parameter set $\tau_i$ whose number of trainable parameters is much fewer than $\theta_i$. Therefore, the memory required by FedDecomp is slightly higher than that of FedAvg.

**Computation cost:** In one round, FedAvg updates $\theta_i$ for $E$ local epochs. According to Eq. (5) and Eq. (6), FedDecomp updates $\tau_i$ for $E_{\text{lora}}$ epochs and update $\sigma_i$ for $E - E_{\text{lora}}$ epochs. Because $\tau_i$ has fewer trainable parameters than $\theta_i$, the computation cost of FedDecomp is lower than that of FedAvg.

**Communication cost:** In one communication round, FedAvg needs to upload $\theta_i$ while FedDecomp needs to upload $\sigma_i$. As $\sigma_i$ has the same number of trainable parameters as $\theta_i$, FedDecomp has the same communication cost as FedAvg.

## 4 EXPERIMENTS

### 4.1 Experiment Setup

**Dataset.** Our main experiments are conducted on three datasets: CIFAR-10 [19], CIFAR-100 [18], and Tiny ImageNet [21]. Experiments on larger datasets involving text modalities are included in the supplemental material. To evaluate the effectiveness of our approach in various scenarios, we adopt the Dirichlet non-IID setting,

a commonly used framework in current FL research [12, 26, 35]. In this setup, each client's data is generated from a Dirichlet distribution represented as $Dir(\alpha)$. As the value of $\alpha$ increases, the level of class imbalance in each client's dataset gradually decreases. Consequently, the Dirichlet non-IID setting allows us to test the performance of our methods across a wide range of diverse non-IID scenarios. For a more intuitive understanding of this concept, we offer a visualization of the data partitioning in the supplemental material.

**Baseline methods.** To verify the efficacy of FedDecomp, we compare it with eight state-of-the-art (SOTA) methods: FedAMP [14], FedRep [7], FedBN [24], FedPer [2], FedRoD [6], pFedSD [17], pFedGate [5], and FedCAC [34]. Among these methods, FedAMP forces clients with similar data distributions to learn from each other. FedBN, FedPer, FedRep, FedRoD, and FedCAC are parameter-partition-based methods that partially personalize parameters. pFedSD and pFedGate are fine-tuning-based methods, whose goal is to adapt the global model to the client's local data. These methods cover the latest advancements in various directions of PFL.

**Selection for hyper-parameters.** We utilize the hyper-parameters specified in the respective papers for each SOTA method. For the FL general hyper-parameters, we set the client number $N = 40$, the local update epochs $E = 5$, the batch size $B = 100$, and the total communication round $T = 300$. Each client is assigned 500 training samples and 100 test samples with the same data distribution. We select the best mean accuracy across all clients as the performance metric. Each experiment is repeated using three seeds, and the mean and standard deviation are reported. We adopt the ResNet [11] network structure. Specifically, we utilize ResNet-8 for CIFAR-10 and ResNet-10 for CIFAR-100 and Tiny ImageNet. In FedDecomp, we adopt the SGD optimizer with learning rate equals 0.1.

### 4.2 Comparison with SOTA methods

In this section, we compare our FedDecomp with several SOTA methods. To ensure a comprehensive evaluation, we consider three different non-IID degrees (i.e., $\alpha \in \{0.1, 0.5, 1.0\}$) on CIFAR-10, CIFAR-100, and Tiny Imagenet.

The results in Table 1 demonstrate that the performance of FedAMP is comparable to other SOTA methods on the CIFAR-10 dataset, but experiences a notable decline on CIFAR-100 and Tiny Imagenet. This is primarily because of its limited capacity to leverage collaboration among clients with diverse data distributions. In contrast, mainstream model partition methods such as FedRep, FedBN, FedPer, FedRoD, and FedCAC enhance collaboration among clients by personalizing parameters sensitive to non-IID data while sharing others. Among these methods, FedRoD distinguishes itself by introducing a balanced global classifier to facilitate comprehensive knowledge exchange, underscoring the potential for improvements in client collaboration within current model partition strategies. On the other hand, fine-tuning-based approaches like pFedSD and pFedGate enable all clients to collaboratively train a global model, fostering extensive knowledge exchange. However, this approach can lead to performance degradation in certain non-IID scenarios due to mutual interference during joint training.

**Table 1: Comparison results under Dirichlet non-IID on CIFAR-10, CIFAR-100, and Tiny Imagenet.**

| Methods | CIFAR-10 | | | CIFAR-100 | | | Tiny Imagenet | | |
|---|---|---|---|---|---|---|---|---|---|
| | 0.1 | 0.5 | 1.0 | 0.1 | 0.5 | 1.0 | 0.1 | 0.5 | 1.0 |
| FedAvg | 60.39±1.46 | 60.41±1.36 | 60.91±0.72 | 34.91±0.86 | 32.78±0.23 | 33.94±0.39 | 21.26±1.28 | 20.32±0.91 | 17.20±0.54 |
| Local | 81.91±3.09 | 60.15±0.86 | 52.24±0.41 | 47.61±0.96 | 22.65±0.51 | 18.76±0.63 | 24.07±0.62 | 8.75±0.30 | 6.87±0.28 |
| FedAMP | 84.99±1.82 | 68.26±0.79 | 64.87±0.95 | 46.68±1.06 | 24.74±0.58 | 18.22±0.41 | 27.85±0.71 | 10.70±0.32 | 7.13±0.21 |
| FedRep | 84.59±1.58 | 67.69±0.86 | 60.52±0.72 | 51.25±1.37 | 26.97±0.33 | 20.63±0.42 | 30.83±1.05 | 12.14±0.28 | 8.37±0.25 |
| FedPer | 84.43±0.47 | 68.80±0.49 | 64.92±0.66 | 51.38±0.94 | 28.25±1.03 | 21.53±0.50 | 32.33±0.31 | 12.69±0.42 | 8.67±0.40 |
| FedBN | 83.55±2.32 | 66.79±1.08 | 62.20±0.67 | 54.35±0.63 | 36.94±0.94 | 33.67±0.12 | 33.34±0.71 | 19.61±0.35 | 16.57±0.44 |
| FedRoD | 86.23±2.12 | 72.34±1.77 | 68.45±1.94 | 60.17±0.48 | 39.88±1.18 | 36.80±0.56 | 41.06±0.77 | 25.63±1.11 | 22.32±1.13 |
| pFedSD | 86.34±2.61 | 71.97±2.07 | 67.21±1.89 | 54.14±0.77 | 41.06±0.83 | 38.27±0.20 | 39.31±0.19 | 19.25±1.80 | 15.91±0.33 |
| pFedGate | **87.25±1.91** | 71.98±1.61 | 67.85±0.87 | 48.54±0.39 | 27.47±0.79 | 22.98±0.03 | 37.59±0.39 | 24.09±0.67 | 19.69±0.14 |
| FedCAC | 86.82±1.18 | 69.83±0.46 | 65.39±0.51 | 57.22±1.52 | 38.64±0.63 | 32.59±0.32 | 40.19±1.20 | 23.70±0.28 | 18.58±0.62 |
| FedDecomp | 85.47±2.06 | **72.78±1.23** | **69.09±1.14** | **63.65±0.53** | **45.96±1.19** | **42.98±0.64** | **44.22±0.55** | **28.25±1.24** | **25.55±0.13** |

Notably, FedDecomp significantly outperforms all baseline methods in the majority of scenarios, particularly as $\alpha$ increases. FedDecomp achieves this by effectively decoupling general and client-specific knowledge through parameter decomposition and mitigating the impact of non-IID through alternating training of full-rank and low-rank matrices.

### 4.3 Ablation Studies

**Effect of $R_l$ and $R_c$.** As we discuss in Section 3.3, $R_l$ and $R_c$ individually denote the ratio of the low-rank matrix's rank to the full-rank matrix's rank in convolutional and fully-connected layers, respectively. They are two important hyper-parameters to control the learning ability of the low-rank matrices. In this section, we evaluate the effect of $R_l$ and $R_c$ on model accuracy. We choose $R_l$ and $R_c$ from {20%, 40%, 60%, 80%, 100%}.

The experimental results are presented in Table 2. Firstly, we observe that the optimal combinations of $(R_c, R_l)$ are (60%, 60%) for CIFAR-10, (80%, 40%) for CIFAR-100, (80%, 40%) for Tiny Imagenet. This underscores the importance of setting the personalized parameter matrices to low rank. Secondly, regarding the optimal combination as the focal point, model accuracy gradually decreases as the rank increases. This occurs because, after this point, the personalized matrices gain more learning capacity and begin to acquire some of the general knowledge. As a result, collaboration among clients on the shared matrices diminishes. As the rank decreases, model accuracy also gradually declines. This is because the personalized matrices fail to capture sufficient client-specific knowledge. This aligns with our expectations. Thirdly, experimental results highlight that model accuracy is more sensitive to changes in $R_l$ than $R_c$. This suggests that the acquisition of client-specific knowledge has a stronger correlation with the classifier than the feature extractor, consistent with prior research such as FedPer, FedRep, and FedRoD.

**Effect of $E_{\text{lora}}$ and $E_{\text{global}}$.** In this section, we verify the effect of $E_{\text{lora}}$ and $E_{\text{global}}$ on model accuracy. For simplicity, we set $E_{\text{global}} = E - E_{\text{lora}}$ and only adjust the value of $E_{\text{lora}}$. We conduct experiments on three datasets under Dirichlet non-IID with $\alpha = 0.1$ and sample $E_{\text{lora}} \in [0, E]$.

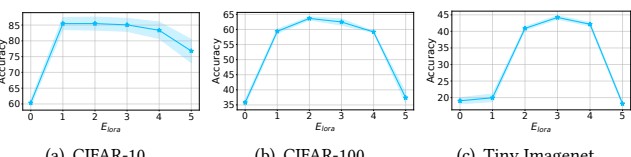

(a) CIFAR-10  (b) CIFAR-100  (c) Tiny Imagenet

**Figure 5: Effect of $E_{\text{lora}}$ in Dirichlet non-IID scenario with $\alpha = 0.1$.**

The experimental results are depicted in Fig. 5. When the $E_{\text{lora}} = 0$, FedDecomp essentially degenerates to FedAvg, and the accuracy closely resembles the FedAvg accuracy presented in Table 1, as expected. As $E_{\text{lora}}$ increases, the accuracy initially rises and then declines. When $E_{\text{lora}} = 5$, FedDecomp degenerates to local training with low-rank parameter matrices. However, due to the constraints imposed by these low-rank matrices on the model's learning capacity, FedDecomp performs less effectively compared to the Local as shown in Table 1.

**Effect of Alternating training.** As we discussed in Section 3.4, different from previous work that trains personalized and shared components simultaneously, we propose to train the personalized part first and then the global part to reduce the impact of non-IID and better extract general knowledge. To evaluate this idea, in this experiment, we compare the performance of two training methods.

The experimental results on three datasets are shown in Table 3. We can see that when the learning task is simple (e.g., a 10-classification task on CIFAR-10), the performance of alternating training and simultaneous training of two matrices is similar. As the learning task becomes increasingly difficult, the performance improvement brought about by alternating training becomes more apparent. This is because, in the case of a simple learning task, the variations in tasks among clients are relatively minor, which facilitates the extraction of general knowledge. However, as the learning task complexity increases, the differences in tasks among clients gradually expand, rendering the extraction of general knowledge more susceptible to non-IID effects. In such scenarios, the utilization

**Table 2: The effect of $R_l$ and $R_c$ on CIFAR-10, CIFAR-100, and Tiny Imagenet under Dirichlet non-IID with $\alpha = 0.1$.**

| Dataset | $R_c$ / $R_l$ | 20% | 40% | 60% | 80% | 100% |
|---|---|---|---|---|---|---|
| CIFAR-10 | 20% | 84.72 ± 2.07 | 84.97 ± 1.74 | 84.73 ± 2.33 | 84.80 ± 2.03 | 84.99 ± 2.19 |
| | 40% | 84.84 ± 2.19 | 84.96 ± 1.87 | 85.27 ± 2.04 | 84.97 ± 1.86 | 85.39 ± 2.01 |
| | 60% | 84.92 ± 1.90 | 85.35 ± 1.96 | **85.47 ± 2.06** | 85.07 ± 2.26 | 85.38 ± 1.76 |
| | 80% | 84.70 ± 1.98 | 85.05 ± 1.66 | 85.25 ± 2.00 | 85.01 ± 1.90 | 85.13 ± 1.95 |
| | 100% | 85.09 ± 1.95 | 85.23 ± 1.89 | 85.15 ± 1.66 | 84.88 ± 1.77 | 85.21 ± 1.62 |
| CIFAR-100 | 20% | 62.00 ± 0.60 | 62.66 ± 0.37 | 61.99 ± 0.97 | 62.48 ± 0.30 | 62.70 ± 0.83 |
| | 40% | 61.70 ± 0.28 | 62.49 ± 0.77 | 62.70 ± 0.59 | **63.65 ± 0.53** | 62.73 ± 0.60 |
| | 60% | 61.71 ± 0.30 | 62.88 ± 0.33 | 62.46 ± 0.53 | 63.12 ± 0.38 | 63.24 ± 0.66 |
| | 80% | 60.76 ± 0.14 | 62.54 ± 0.56 | 62.74 ± 0.54 | 62.15 ± 0.38 | 62.70 ± 0.57 |
| | 100% | 59.54 ± 0.98 | 60.97 ± 0.35 | 61.86 ± 0.74 | 61.96 ± 0.55 | 62.58 ± 0.51 |
| Tiny Imagenet | 20% | 40.77 ± 0.10 | 42.71 ± 0.59 | 43.27 ± 0.43 | 43.78 ± 0.70 | 43.88 ± 0.16 |
| | 40% | 40.14 ± 0.36 | 42.74 ± 0.46 | 43.82 ± 0.46 | **44.22 ± 0.55** | 43.72 ± 0.24 |
| | 60% | 39.39 ± 0.26 | 42.75 ± 0.46 | 43.44 ± 0.43 | 43.85 ± 0.90 | 44.16 ± 0.48 |
| | 80% | 36.90 ± 0.30 | 41.94 ± 0.29 | 42.75 ± 0.49 | 43.21 ± 0.41 | 43.28 ± 0.36 |
| | 100% | 33.90 ± 0.91 | 40.55 ± 0.15 | 41.75 ± 0.81 | 42.16 ± 0.41 | 42.75 ± 0.31 |

**Table 3: The effect of alternating training in FedDecomp on three datasets.**

| Methods | CIFAR-10 | CIFAR-100 | Tiny |
|---|---|---|---|
| Simultaneously | 85.45±1.83 | 61.18±1.05 | 35.37±0.71 |
| Alternatingly | **85.47±2.06** | **63.65±0.53** | **44.22±0.55** |

of our proposed alternating training method becomes increasingly crucial.

**Effect of Model Capacity.** In FedDecomp, we employ an additive decomposition technique on the model. In theory, this approach does not change the model's capacity. However, in practical implementation, the decomposed model introduces low-rank matrices, thereby increasing the number of trainable parameters. This raises questions about whether the decomposed model genuinely enhances the model's capacity and whether the observed performance improvement is primarily a result of the increased number

**Table 4: The effect of low-rank matrices on model capacity.**

| Methods | CIFAR-10 & ResNet-8 | CIFAR-100 & ResNet-10 |
|---|---|---|
| Local | 81.91 ± 3.09 | 47.61 ± 0.96 |
| Local w/ Low-Rank | 81.97 ± 2.62 | 47.64 ± 0.79 |
| FedAvg | 60.39 ± 1.46 | 34.91 ± 0.86 |
| FedAvg w/ Low-Rank | 60.91 ± 0.53 | 35.91 ± 0.70 |

**Table 5: The effect of personalizing low-rank matrices while sharing full-rank matrices on CIFAR-100.**

| Methods | $\alpha = 0.1$ | $\alpha = 0.5$ | $\alpha = 1.0$ |
|---|---|---|---|
| FedDecomp | 63.65±0.53 | 45.96±1.19 | 42.98±0.64 |
| FedDecomp_Reverse | 48.80±0.88 | 23.85±0.99 | 18.52±0.86 |

of trainable parameters. To address these concerns, we conducted an experiment to assess the impact on model capacity.

We conduct experiments using two configurations: CIFAR-10 with the ResNet-8 model and CIFAR-100 with the ResNet-10 model. We established two controlled scenarios: 1) 'Local' and 'Local w/ Low-Rank' indicate models without and with low-rank matrices that are exclusively trained locally. 2) 'FedAvg' and 'FedAvg w/ Low-Rank' indicate models without and with low-rank matrices trained using the FedAvg algorithm. The experimental results are shown in Table 4. Notably, we observe that, in comparison to the original model, the model enhanced with low-rank matrices exhibits only minimal performance improvement. This outcome underscores that our utilization of parameter decomposition does not bring about significant alterations to the model's capacity. Hence, the performance gains achieved by FedDecomp are not solely attributed to modifications in the model itself.

**Effect of Personalize Low-rank Matrices and Sharing Full-rank Matrices.** As discussed in section 3.3, the shared parameters require a high capacity to maintain general knowledge among clients, while personalized parameters only need to preserve client-specific knowledge as a supplement to the general knowledge. To validate this intuition, we evaluate 'FedDecomp_Reverse' which shares low-rank matrices and personalizes full-rank matrices. Other training strategy is the same as FedDecomp and the comparison results on CIFAR-100 are shown in Table 5.

We can see that the performance of 'FedDecomp_Reverse' is not as good as FedDecomp. Moreover, combining the results in Table 1, we can see that the performance of 'FedDecomp_Reverse' is similar to local training. This indicates that in 'FedDecomp_Reverse', the knowledge is mainly preserved in personalized parameters and the knowledge sharing among clients is very limited. This supports our intuition of personalizing low-rank matrices and sharing full-rank matrices.

## 4.4 The Effect of Alternating Training on Model Difference

As we discussed in Section 3.4 and Fig. 4, the primary objective of alternating training is to mitigate the impact of data heterogeneity on

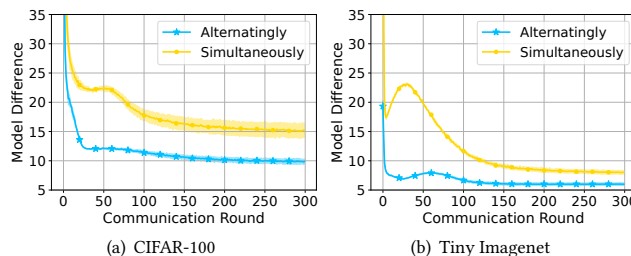

(a) CIFAR-100      (b) Tiny Imagenet

**Figure 6: Effect of training logic on average model difference of $\sigma_i, 1 \leq i \leq N$ and $\overline{\sigma}$ in Dirichlet non-IID scenario with $\alpha = 0.1$.**

**Table 6: The relationship between $\Delta\sigma$ and $\Delta\tau$ when the $\sigma$ is initialized by pre-trained weights.**

| $E_{\text{lora}}$ | 0 | 1 | 2 | 3 | 4 |
|---|---|---|---|---|---|
| Accuracy | 39.35 | 50.37 | 71.47 | 72.00 | **72.42** |
| $\Delta\sigma$ | 37.99 | 31.59 | 22.12 | 16.16 | 10.57 |
| $\Delta\tau$ | 0.00 | 26.48 | 89.68 | 98.26 | 99.63 |

the shared parameters, essentially reducing the deviation of shared parameters to the local minimum point of the client. Consequently, employing alternating training should lead to a reduction in the discrepancies among shared parameters across clients during their local training phases. To validate the effectiveness of alternating training in achieving this goal, we carry out additional experiments to compare the disparities in shared parameters among clients when using alternating training as opposed to not using it. Specifically, we calculate the average model distance between $\sigma_i, 1 \leq i \leq N$ and $\overline{\sigma}$ by $\frac{1}{N} \sum_i^N ||\sigma_i^t - \overline{\sigma}^t||_2$ in each round $t$. The results are shown in Fig. 6. It is evident from the data that, across both datasets, the utilization of alternating training significantly diminishes the differences in the shared parameters among clients. This is consistent with our intuition and analysis.

### 4.5 Experiments with pre-trained model

In FedDecomp, we suppose that the client-specific knowledge is learned by the low-rank matrices. To validate this assumption, we initialize $\sigma$ with pre-trained weights. In this case, the general knowledge is well extracted. If our idea holds, then the $\tau$ should be trained more to learn client-specific knowledge and $\sigma$ should be trained less (i.e., $\Delta\sigma$ should be much smaller than $\Delta\tau$).

Specifically, We initialize the $\sigma_i, i \in [1, N]$ with ImageNet pre-trained weights and conduct an experiment on the CIFAR-100 dataset in the Dirichlet non-IID scenario with $\alpha = 0.1$. We calculate the $\Delta\sigma$ by $||\overline{\sigma}^T - \overline{\sigma}^1||_2$ and the $\Delta\tau$ by $\frac{1}{N} \sum_{i=1}^N ||\tau_i^T - \tau_i^1||_2$. We control the value of $E_{\text{lora}}$ at different levels (higher $E_{\text{lora}}$ means update $\tau$ more often, and results in larger $\Delta\tau$), the results are summarized in the Table 6.

From the table, we can conclude that when $\Delta\tau$ is larger than $\Delta\sigma$, FedDecomp achieves better results. This indicates that with the pre-trained weights, the $\tau$ should be updated more often than $\sigma$ to learn client-specific knowledge. This aligns with our expectations.

**Table 7: PSNR (dB, ↓) values for privacy evaluation on CIFAR10 in Dirichlet non-IID setting with $\alpha = 0.1$.**

| Methods | FedAvg | FedPer | FedRoD | FedDecomp |
|---|---|---|---|---|
| PSNR_Avg | 13.91 | 12.43 | 12.00 | **11.34** |
| PSNR_Max | 17.11 | 19.52 | 16.99 | **13.55** |

### 4.6 Privacy Analysis

In this section, we analyze the privacy protection capability of the FedDecomp. To this end, we adopt the Deep Leakage from Gradient (DLG) method [9, 43] as the attack scheme. DLG is a common attack against FL, and its main idea is: 1) The attacker steals the gradients calculated by each client using local data; 2) The attacker finds the optimal input through iterative optimization, such that the gradient computed with this input is as close as possible to the actual gradient.

In our specific experimental setup, we choose the CIFAR-10 dataset with 20 clients, and the data distribution of each client follows a Dirichlet distribution with $\alpha = 0.1$. For each algorithm, we assume that the gradients of shared parameters can be obtained by the attacker. At training rounds 10, 20, 30, 40, and 50, we attempt to recover each client's 5 training images using the DLG method. To measure the quality of image recovery, we use the Peak Signal-to-Noise Ratio (PSNR), which is defined as $PSNR = -10 \cdot \log_{10}(\frac{||x-x^*||_2^2}{m \cdot n})$, where $x^*$ is the target image to be recovered, $x$ is the image being optimized for recovery, and $m, n$ are the width and height of the image, respectively. A higher PSNR indicates that the recovered image is closer to the original image, which in turn implies weaker privacy protection by the algorithm. Table 7 shows our experimental results, including the average PSNR and the maximum PSNR of the attack results. The results show that our proposed FedDecomp method achieved lower PSNR compared to FedAvg and other personalized methods. This means that using the FedDecomp method, both on average and in the most extreme cases, better privacy protection can be achieved compared to previous methods. We believe that this is because our method can effectively decouple general knowledge from personalized knowledge and keep personalized knowledge well-preserved in the local low-rank branch. As such, it becomes difficult for the DLG method to recover the original image using only the shared part of the gradients.

## 5 CONCLUSION

In this paper, we propose a new PFL method named FedDecomp. FedDecomp decomposes each model parameter matrix into a shared full-rank matrix and a personalized low-rank matrix. To further enhance the acquisition of general knowledge, we devise a training strategy that prioritizes the training of the low-rank matrix to absorb the influence of non-IID during local training. Our extensive experimental evaluations, conducted across multiple datasets characterized by varying degrees of non-IID, unequivocally demonstrate the superior performance of our FedDecomp method when compared to SOTA methods.

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
