# OpenReview forum: "Decoupling General and Personalized Knowledge in Federated Learning via Additive and Low-rank Decomposition"
_acmmm.org/ACMMM/2024/Conference — MM2024 Poster_

### Official Review · Reviewer_55eR · 2024-05-16

**Rating:** 5
**Confidence:** 3

**Summary:**

In this paper,  the authors introduce FedDecomp, a simple but effective PFL paradigm that employs parameter additive decomposition. Experimental results demonstrate that FedDecomp outperforms state-of-the-art methods up to 4.9%.

**Strengths:**

(1) The paper is easy to follow and well-organized. I enjoy reading this paper.

(2) The methodology is novel and technically sound.

(3) The authors conduct extensive experiments to illustrate the proposed method obtains SOTA performance.

**Limitations:**

(1) I strongly encourage the authors to discuss the convergence analysis (either theoretically or empirically) for the proposed FedDecomp.

(2) I strongly encourage the authors to discuss is it possible that we first train on $\sigma$ then on $\tau$? Could it enhance or deteriorate the model performance?

**Suitability:**

3

---

### Official Review · Reviewer_sasX · 2024-05-22

**Rating:** 2
**Confidence:** 3

**Summary:**

This paper presents a novel approach for Personalized Federated Learning (PFL) that addresses the challenge of effectively separating general and client-specific knowledge within model parameters. This paper utilizes a parameter additive decomposition strategy, splitting each parameter into two components: a shared parameter and a personalized one. This method allows for a clearer distinction between general and client-specific knowledge. Additionally, the adoption of a low-rank matrix for personalized parameters acknowledges that less model capacity is needed for client-specific learning compared to general knowledge. An alternating training strategy is also introduced to enhance performance.

**Strengths:**

1. This method effectively distinguishes between general knowledge and client-specific knowledge more efficiently than traditional parameter partitioning approaches.
2. The paper proposes a new alternating training strategy that further improves the model's performance. This strategy optimizes the model's adaptability to different clients' data distributions, enhancing its applicability and robustness.
3. The authors have tested their method on multiple datasets and compared it across various non-IID data distributions.
4. The figures in the paper are well-drawn and clear.

**Limitations:**

1. It seems that the author didn't proofread the paper very carefully, e.g., the text extends beyond the defined margins on lines 538 and 542.
2. The appendix provided in the supplementary material is single-column, which does not fit the double-column format of ACM MM.
3. No code is provided in the supplementary to reproduce the experiments.
4. The performance improvement is not very high, e.g., only 0.8\% of accuracy improvement on the CIFAR-10 dataset when the non-IID setting is 0.5, and less effective when the non-IID setting is 0.1.
5. The experiments provide a limited amount of information. For example, Table 2 is excessively large relative to the amount of information it conveys, authors can consider changing the table to a figure to save space and include the experiments in the appendix into the main manuscript.

**Suitability:**

2

---

### Official Review · Reviewer_vGu3 · 2024-05-24

**Rating:** 2
**Confidence:** 3

**Summary:**

This paper introduces FedDecomp, a novel method in Personalized Federated Learning (PFL) that decouples general and client-specific knowledge via additive parameter decomposition. Each parameter is split into a shared and a low-rank personalized part, ensuring clearer knowledge separation. An alternating training strategy mitigates the influence of non-IID data. Experiments show that FedDecomp outperforms state-of-the-art methods by up to 4.9%, improving model personalization and client collaboration. This approach offers significant advancements in handling data heterogeneity in federated learning.

**Strengths:**

S1: The layout is well clear, with clearly defined notations and related cost analysis, which are mostly overlooked in existing federated work.
S2: FedDecomp reduces the computational burden while preserving essential client-specific knowledge, enhancing efficiency.

**Limitations:**

W1: Vital relevant work is overlooked, e.g., DBE.
W2: It is only applicable for vision or image data, without the further study for language data.
W3: In terms of the experiment results, the performance on CIFAR-10 dataset reported in Table 3 seems to be significantly lower than the results reported in other related works, e.g., FedRoD.
W4: The convergence impact analysis is deficient for the decomposition model.
W5: The privacy analysis is important, but this presentation is less rigorous as an academic evidence.
W6: the low-rank decomposition method is not novel or has more convincing strength for personalized learning.
W7: the effectiveness of with Low rank(by the results of Table 4) is not evidently improved.
W8: computation efficiency is not empirically validated.


[1] Zhang J, Hua Y, Cao J, et al. Eliminating domain bias for federated learning in representation space[J]. Advances in Neural Information Processing Systems, 2024, 36.

**Suitability:**

2

---

### Official Review · Reviewer_V9uC · 2024-05-26

**Rating:** 4
**Confidence:** 1

**Summary:**

The paper introduces FedDecomp, a novel approach to Personalized Federated Learning (PFL) that addresses the challenge of data heterogeneity across clients.

**Strengths:**

1.The paper proposes FedDecomp, which utilizes parameter additive decomposition to separate general knowledge shared among clients and client-specific knowledge. This is a significant advancement over traditional PFL methods that use parameter partitioning, which struggle to effectively distinguish between the two types of knowledge.

2.The paper presents a new training strategy where the personalized low-rank part is trained first, followed by the shared full-rank part. This order is intended to mitigate the influence of non-IID data and improve the extraction of general knowledge.

3.Experimental results across multiple datasets and varying degrees of data heterogeneity demonstrate that FedDecomp outperforms state-of-the-art methods up to 4.9%.

**Limitations:**

1.There could be a need for a deeper theoretical foundation or analysis of the proposed method.

2.While the paper touches on privacy, a more comprehensive analysis could strengthen the paper.

3.The paper could potentially benefit from discussing how FedDecomp might be applied in real-world scenarios.

**Suitability:**

2

---

### Meta-Review · Area_Chair_9ENN · 2024-07-05

**Recommendation:** Accept (Poster)
**Confidence:** 4

**Metareview:**

This paper studies the personalized federated learning problem. The authors propose to decompose the parameters into two sets with one representing shared knowledge and the other representing local knowledge. The proposed solution is interesting. The evaluation is comprehensive, especially considering the results from the rebuttal. The performance improvement is relatively significant. Therefore, after carefully reviewing the reviews and rebuttals, the AC thinks that, despite lacking of theoretical analysis, the authors addressed most of the points identified by the reviewers, and suggests acceptance (poster). Please seriously consider the comments and carefully incorporate them into the final version.